# A DIFFERENTIAL EQUATION APPROACH FOR WASSERSTEIN GANS AND BEYOND

## ABSTRACT

This paper proposes a new theoretical lens to view Wasserstein generative adversarial networks (WGANs). To minimize the Wasserstein-1 distance between the true data distribution and our estimate of it, we derive a distribution-dependent ordinary differential equation (ODE), which represents the gradient flow of the Wasserstein-1 loss, and show that a forward Euler discretization of the ODE converges. This inspires a new class of generative models that naturally integrates persistent training (which we call W1-FE). When persistent training is turned off, we prove that W1-FE reduces to WGAN. When we intensify persistent training appropriately, W1-FE is shown to outperform WGAN in training experiments from low to high dimensions, in terms of both convergence speed and training results.

## 1 INTRODUCTION

Recently, Huang and Zhang (2023) have shown that the original generative adversarial network (GAN) algorithm (Goodfellow et al., 2014) in fact follows the dynamics of an ordinary differential equation (ODE), which represents the gradient flow induced by the Jensen-Shannon divergence (Huang and Zhang, 2023, Proposition 8). Huang and Malik (2024) characterize the gradient flow induced by the Wasserstein-2 distance and propose a class of generative models, called W2-FE, that follows the dynamics of the corresponding ODE (Huang and Malik, 2024, Algorithm 1). Notably, while W2-FE covers W2-GAN in Leygonie et al. (2019) as a special case, it allows for a modification to the generator training that may lead to substantial numerical improvements over W2-GAN (Huang and Malik, 2024, Section 7).

This paper is motivated by the question: can we obtain analogous results for Wasserstein generative adversarial networks (WGANs)? As a mainstream class of GAN algorithms known for its enhanced stability, WGAN (Arjovsky et al., 2017; Gulrajani et al., 2017; Petzka et al., 2018) attempts to generate samples whose distribution is similar to the true data distribution (measured by the Wasserstein-1 distance) using adversarial training motivated by Goodfellow et al. (2014). It is natural to expect that, by slightly modifying the arguments in Huang and Malik (2024), one can similarly derive an ODE that represents the gradient flow induced by the Wasserstein-1 distance and correspondingly propose a new class of generative models that covers and potentially improves WGAN.

Mathematical challenges abound, however, under the the Wasserstein-1 distance. First and foremost, while subdifferential calculus is full-fledged under the Wasserstein-$p$ distance for all $p > 1$ (Ambrosio et al., 2008, Chapter 10), the same construction breaks down when $p = 1$. In particular, "Fréchet differential" (Ambrosio et al., 2008, Corollary 10.2.7) or "Wasserstein gradient" (Carmona and Delarue, 2018, Definition 5.62) is no longer well-defined for $p = 1$. Simply put, it is not even clear how "gradient" should be defined under the Wasserstein-1 distance. On top of this, when showing that a discretization of the gradient-flow ODE converges (as the time step tends to 0), Huang and Malik (2024) crucially rely on an interpolation result from optimal transport: the most cost-efficient path to move one probability measure to another (measured by the Wasserstein-$p$ distance for any $p > 1$) can be interpolated recursively into smaller segments between intermediate measures (Ambrosio et al., 2008, Lemma 7.2.1), where each segment is most cost-efficient in itself and represents one time step in the ODE discretization. Such an interpolation result, again, fails when $p = 1$.

To overcome these challenges, we first recall "linear functional derivative" from the mean field game literature (Carmona and Delarue, 2018) and observe that the "Euclidean gradient of a linear functional derivative" can generally serve as the proper gradient notion in the space of probability measures,

independently of any subdifferential calculus; see the discussion below Definition 3.1. Our first main theoretical result, Proposition 3.1, shows that the linear function derivative of the Wasserstein-1 distance exists and it coincides with the Kantorovich potential in optimal transport (i.e., a 1-Lipschitz function that maximizes the duality formula of the Wasserstein-1 distance; see Definition 2.1). The gradient flow induced by the Wasserstein-1 distance thus takes the form of an ODE that evolves along the negative Euclidean gradient of the Kantorovich potential (i.e., (3.6) below) and a forward Euler discretization can be correspondingly devised (i.e., (4.1) below). As mentioned above, the interpolation result in Ambrosio et al. (2008) can no longer be used to show the convergence of the discretization. Our second main theoretical result, Theorem 4.1, instead relies on the uniform boundedness of the Euclidean gradient of the Kantorovich potential (see Remark 2.1), which is unique to the $p = 1$ case. Such boundedness allows us to show appropriate compactness and equicontinuity of the flow of measures induced by the discretization, so that a refined Arzela-Ascoli argument can be applied to give the convergence of the discretization; see the discussion below Theorem 4.1.

Algorithm 1 (called W1-FE) is designed to simulate the discretization of the gradient-flow ODE. It first computes an estimate of the Kantorovich potential, by following the discriminator training in a typical WGAN algorithm. With the Kantorovich potential estimated, a generator is trained to move along the ODE discretization. As the generator's task is to learn the new distribution at the next time point, *persistent training* techniques (Fischetti et al., 2018) can be naturally incorporated—namely, given a set of samples from the new distribution, the generator's loss descends $K \in \mathbb{N}$ consecutive times to better represent these samples (and thus the new distribution). For the case $K = 1$ (which means no persistent training), Proposition 4.1 shows that W1-FE reduces to WGAN. It is interesting to note that, despite the coincidence under $K = 1$, W1-FE and WGAN are fundamentally different. Specifically, if we also incorporate persistent training into WGAN, the generator updates in W1-FE and WGAN can be quite different under any $K > 1$. That is, $K = 1$ is the only case where they agree; see the discussion below Proposition 4.1 and Remark 4.2 for details.

We train W1-FE with diverse persistency levels $K = 1, 3, 5, 10$ in three experiments that involve datasets from low to high dimensions, including synthetic two-dimensional mixtures of Gaussians and real datasets of USPS, MNIST, and CIFAR-10. Across the three experiments, W1-FE with $K > 1$ converges significantly faster and achieves better training results than the baseline $K = 1$ case (which is WGAN). Intriguingly, while a larger $K$ generally implies faster convergence, we observe numerically a threshold of $K$ beyond which the training results start to deteriorate, possibly due to overfitting. This suggests that taking $K$ to be at that threshold can likely best balance the benefits of persistent training against potential overfitting; see the last paragraph in Section 6 for details.

Many recently developed generative models also feature "Wasserstein gradient flows" (e.g., Fan et al. (2022), Choi et al. (2024), Zhang and Katsoulakis (2023), and Onken et al. (2020)), and the common theme is to minimize a loss function under the geometry induced by the Wasserstein-2 distance. This means that "Wasserstein gradient flows" in the literature should be more precisely referred to as "Wasserstein-2 gradient flows." In other words, simply because subdifferential calculus under the Wasserstein-2 distance and the resulting gradient flows are widely studied (Ambrosio et al., 2008, Chapter 10 and Section 11.2), much of the recent literature has leveraged on this to design new generative models. Our study is distinct from all this, as we right away tackle Wasserstein-1 gradient flows, which are much less understood. As explained before, this focus on Wasserstein-1 gradient flows allows us to recover and even improve WGAN, which by construction minimizes the Wasserstein-1 distance and cannot be easily analyzed by the standard Wasserstein-2 framework.

The rest of this paper is organized as follows. Section 2 introduces the mathematical framework and notation to be used. Section 3 discusses how the "gradient descent" idea can be applied to minimizing the Wasserstein-1 distance and formulates the corresponding gradient-flow ODE. Section 4 defines a forward Euler discretization of the ODE and designs an algorithm (i.e., W1-FE) to simulate this discretization. Section 5 trains W1-FE in three experiments and demonstrates its superiority over WGAN. Section 6 discusses the limitations of our study on both the theoretical and numerical sides. Section 7 concludes our findings.

## 2 MATHEMATICAL PRELIMINARIES

Fix $d \in \mathbb{N}$ and let $\mathcal{L}^d$ be the Lebesgue measure on $\mathbb{R}^d$. Let $\mathcal{P}(\mathbb{R}^d)$ be the set of probability measures on $\mathbb{R}^d$ and $\mathcal{P}_p(\mathbb{R}^d)$, for $p \geq 1$, be the set of elements in $\mathcal{P}(\mathbb{R}^d)$ with finite $p^{th}$ moments, i.e.,

$$\mathcal{P}_p(\mathbb{R}^d) := \left\{ \mu \in \mathcal{P}(\mathbb{R}^d) : \int_{\mathbb{R}^d} |y|^p d\mu(y) < \infty \right\}.$$

The Wasserstein-$p$ distance, a metric on $\mathcal{P}_p(\mathbb{R}^d)$, is defined by

$$W_p(\mu, \nu) := \left( \inf_{\gamma \in \Gamma(\mu, \nu)} \int_{\mathbb{R}^d \times \mathbb{R}^d} |x - y|^p \, d\gamma(x, y) \right)^{1/p}, \quad \forall \mu, \nu \in \mathcal{P}_p(\mathbb{R}^d), \tag{2.1}$$

where $\Gamma(\mu, \nu)$ is the set of all probability measures on $\mathbb{R}^d \times \mathbb{R}^d$ whose marginals on the first and second coordinates are $\mu$ and $\nu$, respectively (Villani, 2009, Definition 6.1). For $p = 1$, we recall the Kantorovich-Rubinstein duality formula for the $W_1$ distance (Villani, 2009, (5.11)), i.e.,

$$W_1(\mu, \nu) \quad = \quad \sup_{\varphi : \mathbb{R}^d \to \mathbb{R}, \, ||\varphi||_{\text{Lip}} \leq 1} \left\{ \int_{\mathbb{R}^d} \varphi \, d\mu - \int_{\mathbb{R}^d} \varphi \, d\nu \right\}, \tag{2.2}$$

where "$||\varphi||_{\text{Lip}} \leq 1$" means that $\varphi : \mathbb{R}^d \to \mathbb{R}$ is a 1-Lipschitz function.

**Definition 2.1.** *A 1-Lipschitz $\varphi : \mathbb{R}^d \to \mathbb{R}$ that maximizes (2.2) is called a (maximal) Kantorovich potential from $\mu$ to $\nu$ and will be denoted by $\varphi_\mu^\nu$ to emphasize its dependence on $\mu$ and $\nu$.*

The general definition of a (maximal) Kantorovich potential is stated for any $p \geq 1$; see the remark above Ambrosio et al. (2008, Theorem 6.15). For $p = 1$, it reduces specifically to Definition 2.1, thanks to the discussion in Villani (2009, Particular Case 5.4).

**Remark 2.1.** *For any $\mu, \nu \in \mathcal{P}_1(\mathbb{R}^d)$, by Villani (2009, Theorem 5.10 (iii)), a Kantorivich potential $\varphi_\mu^\nu$ generally exists. As $\varphi_\mu^\nu$ is 1-Lipschitz, $\nabla \varphi_\mu^\nu(x)$ exists with $|\nabla \varphi_\mu^\nu(x)| \leq 1$ for $\mathcal{L}^d$-a.e. $x \in \mathbb{R}^d$.*

## 3 PROBLEM FORMULATION

Let $\mu_{\mathrm{d}} \in \mathcal{P}_1(\mathbb{R}^d)$ denote the (unknown) data distribution. Starting with an arbitrary initial estimate $\mu_0 \in \mathcal{P}_1(\mathbb{R}^d)$ of $\mu_{\mathrm{d}}$, we aim to improve our estimate progressively and ultimately solve the problem

$$\min_{\mu \in \mathcal{P}_1(\mathbb{R}^d)} W_1(\mu, \mu_{\mathrm{d}}). \tag{3.1}$$

As it can be checked directly that $\mu \mapsto W_1(\mu, \mu_{\mathrm{d}})$ is convex on $\mathcal{P}_1(\mathbb{R}^d)$ (Appendix A.1), it is natural to ask if (3.1) can be solved by gradient descent, the traditional wisdom of convex minimization. Recall that for a convex $f : \mathbb{R}^d \to \mathbb{R}$, if its minimizer $y^* \in \mathbb{R}^d$ exists, it can be found by gradient descent in $\mathbb{R}^d$. Specifically, for any initial point $y \in \mathbb{R}^d$, the ODE

$$dY_t = -\nabla f(Y_t)dt, \quad Y_0 = y \in \mathbb{R}^d \tag{3.2}$$

converges to $y^*$ as $t \to \infty$. For the derivation of a similar gradient-descent ODE for (3.1), where the minimizer is clearly $\mu_{\mathrm{d}} \in \mathcal{P}_1(\mathbb{R}^d)$, the crucial question is how the "gradient" of the function

$$J(\mu) := W_1(\mu, \mu_{\mathrm{d}}), \quad \mu \in \mathcal{P}_1(\mathbb{R}^d) \tag{3.3}$$

should be defined. As mentioned in the introduction, subdifferential calculus is well-developed in $\mathcal{P}_p(\mathbb{R}^d)$ for all $p > 1$ (Ambrosio et al., 2008, Chapter 10), but the same construction breaks down exactly when $p = 1$. As a result, neither "Fréchet differential" (Ambrosio et al., 2008, Corollary 10.2.7) nor the equivalent "Wasserstein gradient" (Carmona and Delarue, 2018, Definition 5.62) is well-defined in $\mathcal{P}_1(\mathbb{R}^d)$. To circumvent this, let us first recall "linear functional derivative" from the mean field game literature.

**Definition 3.1.** *Let $\mathcal{S} \subseteq \mathcal{P}(\mathbb{R}^d)$ be convex. A linear functional derivative of $U : \mathcal{S} \to \mathbb{R}$ is a function $\frac{\delta U}{\delta m} : \mathcal{S} \times \mathbb{R}^d \to \mathbb{R}^d$ that satisfies*

$$\lim_{\epsilon \to 0^+} \frac{U(\mu + \epsilon(\nu - \mu)) - U(\mu)}{\epsilon} = \int_{\mathbb{R}^d} \frac{\delta U}{\delta m}(\mu, y) \, d(\nu - \mu)(y), \quad \forall \mu, \nu \in \mathcal{S}. \tag{3.4}$$

The above definition is in line with Jourdain and Tse (2021, Definition 2.1), where $\mathcal{S} = \mathcal{P}_p(\mathbb{R}^d)$ for $p \geq 1$, and Carmona and Delarue (2018, Definition 5.43), where $\mathcal{S} = \mathcal{P}_2(\mathbb{R}^d)$. Note that "$\delta U / \delta m$" is simply a common notation for a function satisfying (3.4), where "$m$" indicates that the variable in discussion is a probability measure and "$\delta/\delta m$" alludes to a kind of differentiation with respect to $m$.

The key observation here is that the "Euclidean gradient of a linear functional derivative," i.e., $\nabla \frac{\delta U}{\delta m}(\mu, \cdot) : \mathbb{R}^d \to \mathbb{R}^d$, can generally serve as the "gradient of $U$ at a measure $\mu$." For $\mathcal{S} = \mathcal{P}_2(\mathbb{R}^d)$, Carmona and Delarue (2018, Proposition 5.48 and Theorem 5.64) show that $\nabla \frac{\delta U}{\delta m}(\mu, \cdot)$ in fact coincides with the Wasserstein gradient of $U$ at $\mu \in \mathcal{P}_2(\mathbb{R}^d)$. For $\mathcal{S} = \mathcal{P}^r(\mathbb{R}^d) := \{\mu \in \mathcal{P}(\mathbb{R}^d) : \mu \ll \mathcal{L}^d, \frac{d\mu}{d\mathcal{L}^d} \in C^1(\mathbb{R}^d)\}$, where Wasserstein gradients are not well-defined, Huang and Zhang (2023) show that $\nabla \frac{\delta U}{\delta m}(\mu, \cdot)$ still fulfills a gradient-type property. Specifically, for any $\mu \in \mathcal{P}^r(\mathbb{R}^d)$ and $\xi : \mathbb{R}^d \to \mathbb{R}^d$, let $\mu_\epsilon^\xi$ be the law of $Y + \epsilon \xi(Y)$, where $Y$ is a random variable whose law is $\mu$. For sufficiently smooth and compactly supported $\xi$, Huang and Zhang (2023, Proposition 5) shows that $\mu_\epsilon^\xi \in \mathcal{P}^r(\mathbb{R}^d)$ and

$$\lim_{\epsilon \to 0^+} \frac{U(\mu_\epsilon^\xi) - U(\mu)}{\epsilon} = \int_{\mathbb{R}^d} \nabla \frac{\delta U}{\delta m}(\mu, y) \cdot \xi(y) d\mu(y),$$

provided that $\frac{\delta U}{\delta m}$ is locally integrable and sufficiently continuous. That is, for any $y \in \mathbb{R}^d$, $\nabla \frac{\delta U}{\delta m}(\mu, y)$ specifies how moving along $\xi(y)$ instantaneously changes the function value from $U(\mu)$, which suggests that $\nabla \frac{\delta U}{\delta m}(\mu, \cdot) : \mathbb{R}^d \to \mathbb{R}^d$ should be the proper "gradient of $U$ at $\mu \in \mathcal{P}^r(\mathbb{R}^d)$."

In view of this, in our case of $\mathcal{S} = \mathcal{P}_1(\mathbb{R}^d)$, where Wasserstein gradients are again not well-defined, we take the "gradient of $J$ in (3.3) at $\mu \in \mathcal{P}_1(\mathbb{R}^d)$" to be $\nabla \frac{\delta J}{\delta m}(\mu, \cdot) : \mathbb{R}^d \to \mathbb{R}^d$. The resulting gradient-descent ODE for (3.1), in analogy to (3.2), is then

$$dY_t = -\nabla \frac{\delta J}{\delta m}(\mu^{Y_t}, Y_t) \, dt, \quad \mu^{Y_0} = \mu_0 \in \mathcal{P}_1(\mathbb{R}^d). \tag{3.5}$$

This ODE, intriguingly, is *distribution-dependent*. At time 0, $Y_0$ is an $\mathbb{R}^d$-valued random variable whose law is $\mu_0 \in \mathcal{P}_1(\mathbb{R}^d)$, an arbitrarily specified initial distribution. This initial randomness trickles through the ODE dynamics in (3.5), such that $Y_t$ remains an $\mathbb{R}^d$-valued random variable, with its law denoted by $\mu^{Y_t} \in \mathcal{P}_1(\mathbb{R}^d)$, at every $t > 0$. The evolution of the ODE is then determined jointly by the "gradient of $J$" at the present distribution $\mu^{Y_t} \in \mathcal{P}_1(\mathbb{R}^d)$ (i.e., the function $\nabla \frac{\delta J}{\delta m}(\mu^{Y_t}, \cdot)$) and the actual realization of $Y_t$ (which is plugged into $\nabla \frac{\delta J}{\delta m}(\mu^{Y_t}, \cdot)$).

To ensure that ODE (3.5) makes sense and is tractable enough, one needs to show that $\frac{\delta J}{\delta m}$ exists and admits a concrete characterization. Our first main theoretic result serves this purpose.

**Proposition 3.1.** *For any $\mu \in \mathcal{P}_1(\mathbb{R}^d)$, a Kantorovich potential $\varphi_\mu^{\mu_\mathrm{d}}$ (Definition 2.1) is a linear functional derivative of $J : \mathcal{P}_1(\mathbb{R}^d) \to \mathbb{R}$ in (3.3) at $\mu \in \mathcal{P}_1(\mathbb{R}^d)$ (Definition 3.1 with $\mathcal{S} = \mathcal{P}_1(\mathbb{R}^d)$). Specifically, for any $\mu \in \mathcal{P}_1(\mathbb{R}^d)$,*

$$\frac{\delta J}{\delta m}(\mu, y) = \varphi_\mu^{\mu_\mathrm{d}}(y) \quad \forall y \in \mathbb{R}^d.$$

The proof of Proposition 3.1 is relegated to Appendix A.2. To the best of our knowledge, Proposition 3.1 is the first result that establishes a precise connection between "Kantorovich potential" in optimal transport and "linear functional derivative" in the mean field game literature.

Thanks to Proposition 3.1, ODE (3.5) now becomes

$$dY_t = -\nabla \varphi_{\mu^{Y_t}}^{\mu_\mathrm{d}}(Y_t) \, dt, \quad \mu^{Y_0} = \mu_0 \in \mathcal{P}_1(\mathbb{R}^d). \tag{3.6}$$

That is, the evolution of the ODE is determined jointly by a Kantorovich potential from the present distribution $\mu^{Y_t}$ to $\mu_\mathrm{d}$ (i.e., the function $\varphi_{\mu^{Y_t}}^{\mu_\mathrm{d}}(\cdot)$) and the actual realization of $Y_t$ (which is plugged into $\nabla \varphi_{\mu^{Y_t}}^{\mu_\mathrm{d}}(\cdot)$).

## 4 A DISCRETIZATION OF ODE (3.6)

Given $\epsilon > 0$ and an initial random variable $Y_{0,\epsilon} = Y_0$ with a given law $\mu^{Y_0} = \mu_0 \in \mathcal{P}_1(\mathbb{R}^d)$, we consider a new random variable defined by

$$Y_{1,\epsilon} := Y_{0,\epsilon} - \epsilon \nabla \varphi_{\mu^{Y_{0,\epsilon}}}^{\mu_\mathrm{d}}(Y_{0,\epsilon}).$$

Note that this is the very first step, from time 0 to time $\epsilon$, in a forward Euler discretization of ODE (3.6). Using the law of $Y_{1,\epsilon}$, denoted by $\mu^{Y_{1,\epsilon}}$, we can obtain a Kantorovich potential $\varphi_{\mu^{Y_{1,\epsilon}}}^{\mu_{\mathrm{d}}}$ from the present distribution $\mu^{Y_{1,\epsilon}}$ at time $\epsilon$ to $\mu_{\mathrm{d}}$. This allows us to perform another forward Euler update and get $Y_{2,\epsilon} := Y_{1,\epsilon} - \epsilon \nabla \varphi_{\mu^{Y_{1,\epsilon}}}^{\mu_{\mathrm{d}}}(Y_{1,\epsilon})$. We may continue this procedure and obtain a sequence of random variables $\{Y_{n,\epsilon}\}_{n \in \mathbb{N}}$, with

$$Y_{n,\epsilon} := Y_{n-1,\epsilon} - \epsilon \nabla \varphi_{\mu^{Y_{n-1,\epsilon}}}^{\mu_{\mathrm{d}}}(Y_{n-1,\epsilon}), \quad \forall n \in \mathbb{N}. \tag{4.1}$$

This discretization recursively defines a sequence of measures $\{\mu^{Y_{n-1,\epsilon}}\}_{n \in \mathbb{N}}$ in $\mathcal{P}_1(\mathbb{R}^d)$. A piecewise constant flow of measures $\mu_\epsilon : [0, \infty) \to \mathcal{P}_1(\mathbb{R}^d)$ can then be defined by

$$\mu_\epsilon(t) := \mu^{Y_{n-1,\epsilon}} \quad \text{for } t \in [(n-1)\epsilon, n\epsilon), \ n \in \mathbb{N} \tag{4.2}$$

Our second main theoretic result, stated below, establishes the convergence of $\mu_\epsilon$ as $\epsilon \to 0^+$. Its proof is relegated to Appendix A.4.

**Theorem 4.1.** *For any $\epsilon > 0$, let $\mu_\epsilon : [0, \infty) \to \mathcal{P}_1(\mathbb{R}^d)$ be defined as in (4.2) and assume that $\mu_\epsilon(t) \ll \mathcal{L}^d$ for all $t \geq 0$. Then, there exists a sequence $\{\epsilon_k\}_{k \in \mathbb{N}}$ with $\epsilon_k \to 0^+$ and a curve $\mu^* : [0, \infty) \to \mathcal{P}_1(\mathbb{R}^d)$ such that*

$$\lim_{k \to \infty} W_1(\mu_{\epsilon_k}(t), \mu^*(t)) = 0 \quad \forall t > 0.$$

*Furthermore, $t \mapsto \mu^*(t)$ is uniformly continuous (in the $W_1$ sense) on compacts of $[0, \infty)$.*

At first glance, one might suspect that Theorem 4.1 is a straightforward extension of Huang and Malik (2024, Theorem 5.2) from $\mathcal{P}_2(\mathbb{R}^d)$ to the larger space $\mathcal{P}_1(\mathbb{R}^d)$. In fact, proving Theorem 4.1 requires completely different arguments. Huang and Malik (2024, Theorem 5.2) is established by an interpolation argument: for any $p > 1$, the most cost-efficient path to move one probability measure to another in $\mathcal{P}_p(\mathbb{R}^d)$, measured by the $W_p$ distance, can be interpolated recursively into smaller segments between intermediate measures, where each segment is most cost-efficient in itself (Ambrosio et al., 2008, Lemma 7.2.1). This allows a suitable ODE discretization to correspond to the smaller segments (Huang and Malik, 2024, Proposition 5.3), thereby admitting a well-defined limit (i.e., the whole most cost-efficient path). The interpolation result in Ambrosio et al. (2008, Lemma 7.2.1), however, does not hold for $p = 1$. Theorem 4.1 instead relies on the uniform boundedness of the gradient of *any* Kantorovich potential (Remark 2.1), which is unique to the $p = 1$ case. Such boundedness allows us to show that $\{\mu_\epsilon(t) : \epsilon > 0, t \in [0, T]\}$ in (4.2) is compact in $\mathcal{P}_1(\mathbb{R}^d)$ for any $T > 0$ and the curve $t \mapsto \mu_\epsilon(t)$ becomes equicontinuous as $\epsilon \to 0^+$, so that a refined Arzela-Ascoli argument can be applied to give the convergence of $\mu_\epsilon$ as $\epsilon \to 0^+$; see Appendix A.4 for details.

**Remark 4.1.** *Theorem 4.1 is important for numerical implementation: it asserts that our discretization scheme (4.1) is stable for small time steps and there is a well-defined limit. Moreover, in light of its proof in Appendix A.4, the established convergence actually holds much more generally. If the discretization (4.1) is modified to*

$$Y_{n,\epsilon} := Y_{n-1,\epsilon} - \epsilon \nabla g_{n-1,\epsilon}(Y_{n-1,\epsilon}), \quad \forall n \in \mathbb{N},$$

*where each $g_{n-1,\epsilon}$ is a 1-Lipschitz function (but not necessarily the Kantorovich potential $\varphi_{\mu^{Y_{n-1,\epsilon}}}^{\mu_{\mathrm{d}}}$), the same convergence result in Theorem 4.1 remains true. This is because the arguments in Appendix A.4 hinge on only the 1-Lipschitz continuity of $g_{n-1,\epsilon}$ (for $|\nabla g_{n-1,\epsilon}|$ to be bounded by 1), but not the specific form of $g_{n-1,\epsilon}$. That is, our discretization scheme (4.1) is robust in the following sense: in actual computation, as long as the estimated $\varphi_{\mu^{Y_{n-1,\epsilon}}}^{\mu_{\mathrm{d}}}$ is 1-Lipschitz (facilitated by the discriminator's regularization in Gulrajani et al. (2017) and Petzka et al. (2018)), the scheme remains stable for small time steps and there is a well-defined limit.*

Thanks to the convergence result in Theorem 4.1, we propose an algorithm (called **W1-FE**) to simulate (4.1); see Algorithm 1. We use two neural networks to carry out the simulation, one for the Kantorovich potential $\varphi : \mathbb{R}^d \to \mathbb{R}$ and the other for the generator $G_\theta : \mathbb{R}^\ell \to \mathbb{R}^d$, where $\mathbb{R}^\ell$ (with $\ell \leq d$) is the latent space where we sample priors. To compute $\varphi$, we can follow the discriminator training in any well-known WGAN algorithm, e.g., vanilla WGAN from Arjovsky et al. (2017), W1-GP from Gulrajani et al. (2017), or W1-LP from Petzka et al. (2018), to obtain an estimate of the Kantorovich potential from the distribution of samples generated by $G_\theta$ to that of the data, $\mu_{\mathrm{d}}$. To

allow such generality in Algorithm 1, we simply denote the computation of $\varphi$ by `SimulatePhi`$(\theta)$ and treat it as a black box function. When we use the method of Petzka et al. (2018) (or Gulrajani et al. (2017)) to compute $\varphi$, Algorithm 1 will be referred to as **W1-FE-LP** (or **W1-FE-GP**).

The generator $G_\theta$ is trained by explicitly following the (discretized) ODE (4.1). We start with a set of priors $\{z_i\}$, produce a sample $y_i = G_\theta(z_i)$ of $\mu^{Y_n,\epsilon}$, and then use a forward Euler step to compute a sample $\zeta_i$ of $\mu^{Y_{n+1},\epsilon}$. The generator's task is to learn how to produce samples indistinguishable from the points $\{\zeta_i\}$—or more precisely, to learn the distribution $\mu^{Y_{n+1},\epsilon}$, represented by the points $\{\zeta_i\}$. To this end, we fix the points $\{\zeta_i\}$ and update the generator $G_\theta$ by descending the mean square error (MSE) between $\{G_\theta(z_i)\}$ and $\{\zeta_i\}$ up to $K \in \mathbb{N}$ times. It is worth noting that throughout the $K$ updates of $G_\theta$, the points $\{\zeta_i\}$ are kept unchanged. This sets us apart from the standard implementation of stochastic gradient descent (SGD), but for a good reason: as our goal is to learn the distribution represented by $\{\zeta_i\}$, it is important to keep $\{\zeta_i\}$ unchanged for the eventual $G_\theta$ to more accurately represent $\mu^{Y_{n+1},\epsilon}$, such that the (discretized) ODE (4.1) is more accurately simulated.

Note that how we update the generator $G_\theta$ corresponds to *persistent training* in Fischetti et al. (2018), a technique that reuses the same minibatch for $K$ consecutive SGD iterations. Experimental results in Fischetti et al. (2018) show that using a *persistency level* of five (i.e., taking $K = 5$) achieves much faster convergence on the CIFAR-10 dataset Fischetti et al. (2018, Figure 1). In our numerical examples (see Section 5), we will also show that increasing the persistency level appropriately can markedly improve training performance.

---

**Algorithm 1** W1-FE, our proposed algorithm.

---

**Require:** Input measures $\mu_0, \mu_{\mathrm{d}}$, batch sizes $m \in \mathbb{N}$, generator learning rate $\gamma_g > 0$, time step $\epsilon > 0$, persistency value $K \in \mathbb{N}$, function `SimulatePhi` to approximate Kantorovich potential, generator $G_\theta$ parameterized as a deep neural network.
    **for** Number of training epochs **do**
        $\varphi \leftarrow$ `SimulatePhi`$(\theta)$                            ▷ Compute Kantorovich potential
        Sample a batch $(z_1, \cdots, z_m)$ of priors
        Compute $y_i \leftarrow G_\theta(z_i)$
        Compute $\zeta_i \leftarrow y_i - \epsilon \nabla\varphi(y_i)$.
        **for** $K$ generator updates **do**                        ▷ Persistent training
            Update $\theta \leftarrow \theta - \frac{\gamma_g}{m}\nabla_\theta \sum_i |\zeta_i - G_\theta(z_i)|^2$.
        **end for**
    **end for**

---

### 4.1 A Comparison: W1-FE and WGAN

Our first finding is that W1-FE actually covers WGAN as a special case. For the case $K = 1$ in Algorithm 1 (i.e., W1-FE), the generator update reduces to the standard SGD without persistent training, which turns Algorithm 1 into standard WGAN algorithms.

**Proposition 4.1.** *The WGAN algorithms presented in Arjovsky et al. (2017), Gulrajani et al. (2017), Petzka et al. (2018) are special cases of Algorithm 1 with $K = 1$.*

*Proof.* Take `SimulatePhi` in Algorithm 1 to be the discriminator update in an aforementioned WGAN algorithm, such that the produced $\varphi$ is exactly the estimated Kantorovich potential in the WGAN algorithm. Then, it suffices to show that the generator update in Algorithm 1, when $K = 1$, coincides with that in the corresponding WGAN algorithm. Observe that with $K = 1$,

$$
\begin{aligned}
\nabla_\theta \frac{1}{m}\sum_{i=1}^m |\zeta_i - G_\theta(z_i)|^2 &= -\frac{2}{m}\sum_{i=1}^m (\zeta_i - G_\theta(z_i))\nabla_\theta G_\theta(z_i) \\
&= \frac{2}{m}\sum_{i=1}^m \epsilon\nabla\varphi(G_\theta(z_i))\nabla_\theta G_\theta(z_i) = \frac{2\epsilon}{m}\nabla_\theta \sum_{i=1}^m \varphi(G_\theta(z_i)),
\end{aligned}
\tag{4.3}
$$

where the second equality follows from $\zeta_i = G_\theta(z_i) - \epsilon\nabla\varphi(G_\theta(z_i))$, due to the two lines above the generator update in Algorithm 1. That is, the generator update in Algorithm 1 is now $\theta \leftarrow \theta - \gamma_g \frac{2\epsilon}{m}\nabla_\theta \sum_{i=1}^m \varphi(G_\theta(z_i))$, the same as that in the WGAN algorithm with a learning rate $2\gamma_g\epsilon$. $\quad\square$

The above result is somewhat unexpected: after all, W1-FE builds upon our gradient-flow approach, distinct from the two-player min-max game perspective that underlies WGAN. Proposition 4.1 shows that the two fundamentally different methods can actually coincide, when the computation of the gradient flow, or the (discretized) ODE (4.1), is *crude*—in the sense that $\mu^{Y_{n+1,\epsilon}}$, the distribution along the ODE at the next time step, is less accurately approximated under $K = 1$.

**Remark 4.2.** *It is tempting to think that if one enforces persistent training also in WGAN (i.e., performs SGD in its generator update $K \in \mathbb{N}$ consecutive times with the same minibatch), WGAN will become our W1-FE. This is however not the case. When $G_\theta$ is updated for the second time in W1-FE, the second equality in (4.3) no longer holds, as "$\zeta_i = G_\theta(z_i) - \epsilon\nabla\varphi(G_\theta(z_i))$" is true only when $G_\theta$ is obtained from the previous iteration and has not been updated yet. The connection between W1-FE and WGAN then breaks down, starting from $K = 2$. That is, even when persistent training is included in WGAN, the generator updates in W1-FE and WGAN coincide only for $K = 1$, and can in general be quite different for $K > 1$.*

## 5 NUMERICAL EXPERIMENTS

This section contains three training experiments with datasets from low to high dimensions. In each experiment, we carry out the training task using W1-FE-LP with diverse persistency levels $K = 1, 3, 5, 10$. The case $K = 1$ can be viewed as the baseline, as Proposition 4.1 indicates that it is equivalent to the refined WGAN algorithm in Petzka et al. (2018) (i.e., W1-LP), which is arguably one of the most well-performing and stable WGAN algorithms.

First, we consider learning a two-dimensional mixture of Gaussians arranged on a circle from an initially given Gaussian distribution (Metz et al., 2017). Figure 1 shows the qualitative evolution of the models, while Figure 2 presents the actual $W_1$ losses. In Figure 2, W1-FE-LP with $K = 3$ and $K = 5$ converges to a similar loss level as the baseline $K = 1$ case (i.e., W1-LP), but achieves it much faster than W1-LP in both number of epochs[1] and wallclock time. These gains are partially lost with $K = 10$, possibly due to overfitting. The computation in Figures 1 and 2 takes 10 discriminator updates per training epoch, $\gamma_g = 10^{-4}$, and minibatches of size $m = 512$, and uses a three-layer perceptron for the generator and discriminator, where each hidden layer contains 128 neurons. All neural networks are trained using the Adam stochastic gradient update rule (Kingma and Ba, 2015). We let $\epsilon = 1$, for $\gamma_g$ is already small and thus controlls any possible overshooting from backpropagation.

Next, we consider domain adaptation from the USPS dataset (Hull, 1994) to the MNIST dataset (Deng, 2012) and evaluate the performance of our algorithms every 100 training epochs using a 1-nearest neighbor (1-NN) classifier. This is the same performance metric as in Seguy et al. (2018), although it was used there only once at the end of training. Figure 3 displays the results. With the persistency level $K = 3$, W1-FE-LP converges significantly faster and achieves consistently a higher accuracy rate than the baseline case $K = 1$ (i.e., W1-LP). Indeed, it takes W1-LP 6000 epochs to attain its ultimate accuracy rate, which is achieved by W1-FE-LP with $K = 3$ by epoch 2000; W1-FE-LP with $K = 3$ continues to improve after epoch 2000, yielding the best accuracy rate among all the models. Raising persistency level to $K = 5$ and $K = 10$ further accelerates the convergence before epoch 2000, but the ultimate accuracy rate achieved worsens slightly, which might result from overfitting. The computation in Figure 3 takes $\gamma_g = 10^{-4}$, $\epsilon = 1$, minibatches of size $m = 64$, and 5 discriminator updates per training epoch.

Finally, we train our algorithms on the CIFAR-10 dataset. The prior distribution (i.e., the input $z$ into the generator in Algorithm 1) is a 100-dimensional standard Gaussian and we transform it through a multi-layer convolutional neural network into an image of dimension $3 \times 32 \times 32$. The performance is evaluated by the Fréchet inception distance (FID) (Heusel et al., 2017), a common criterion for training quality of high-dimensional images. The results are displayed in Figure 4, where the FID is calculated using a pre-trained ResNet18 model on CIFAR-10, publicly available on the Github repository Phan (2021). With a larger persistency level $K$, W1-FE-LP converges faster and consistently achieves a lower FID than the baseline model $K = 1$ (i.e., W1-LP). In particular, it takes W1-LP 30000 epochs to achieve an FID about 8, which is achieved by W1-FE-LP with $K = 10$ by epoch 10000. As an example, the images generated by epoch 500 are far clearer under $K = 10$ than under any smaller $K$ value, as shown in Figure 5. Figure 4 also shows that W1-FE-LP with $K = 5$

---

[1]One "training epoch" refers to learning one Euler step in (4.1), i.e., one iteration of the loop in Algorithm 1.

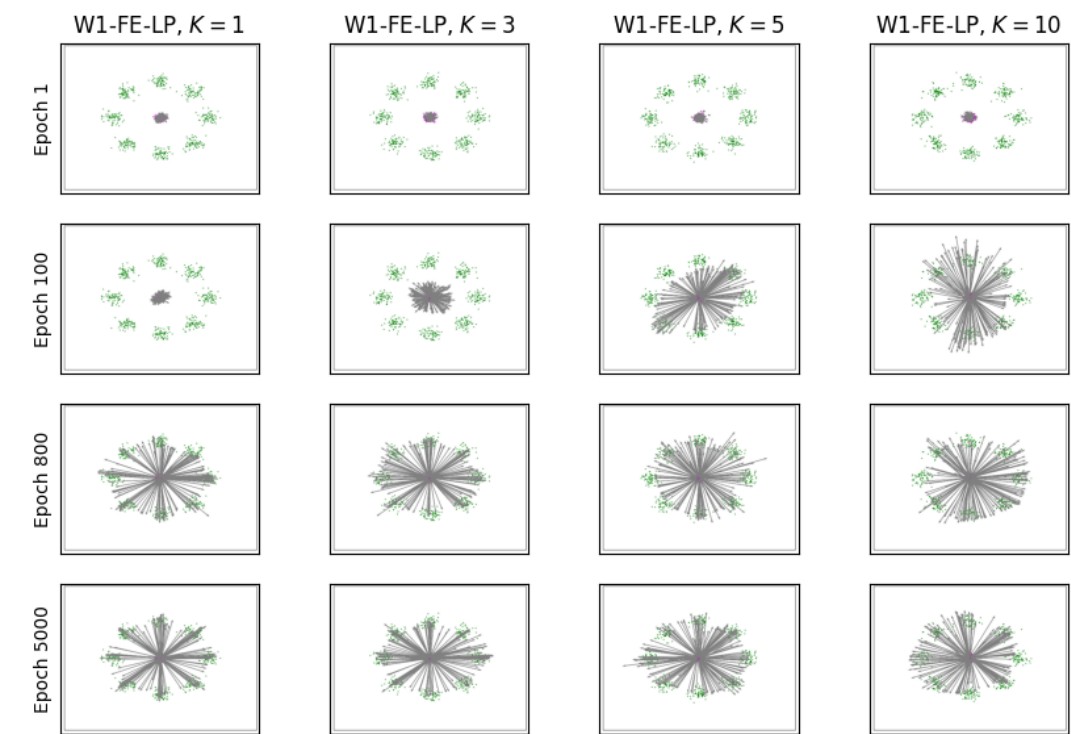

Figure 1: Qualitative evolution of learning process. A sample from the target distribution is given in green, a sample from the initial distribution is in magenta, and the transport rays by the generator are given in the grey arrows. The generate samples lie at the head of each grey arrow.

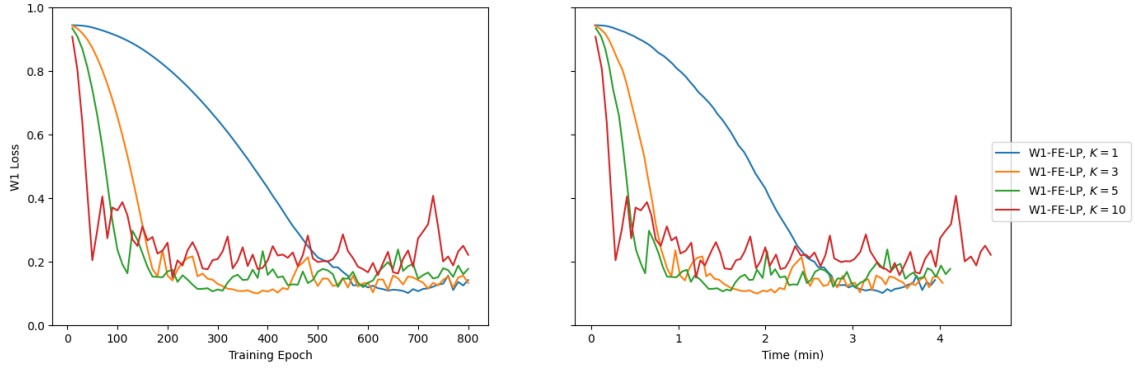

Figure 2: $W_1$ loss of W1-FE-LP with persistency levels $K = 1, 3, 5, 10$ against training epoch (left) and wallclock time (right), respectively.

achieves the smallest FID by epoch 30000. Raising persistency level to $K = 10$ further accelerates the convergence before epoch 20000, but the ultimate FID achieved worsens slightly, possibly due to overfitting. The computation in Figures 4 and 5 takes $\gamma_g = 10^{-4}$, time step $\epsilon = 1$, minibatches of size $m = 64$, and 5 discriminator updates per training epoch.

## 6    LIMITATIONS

While we have made important progress on the theoretical development, several key questions remain open. Recall that Algorithm 1 builds upon (4.1), which is a discretization of the gradient-flow ODE (3.6). While we have shown in Theorem 4.1 that the discretization has a well-defined limit $\mu^*(t)$ in

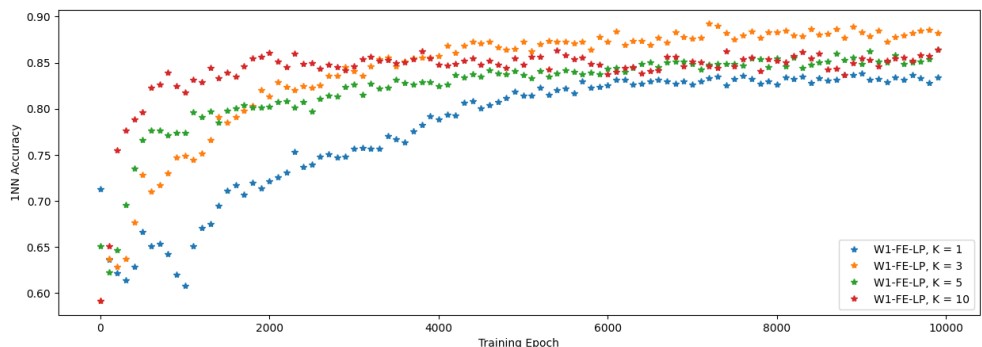

Figure 3: 1-NN classifier accuracy against training epoch for domain adaptation from USPS to MNIST datasets.

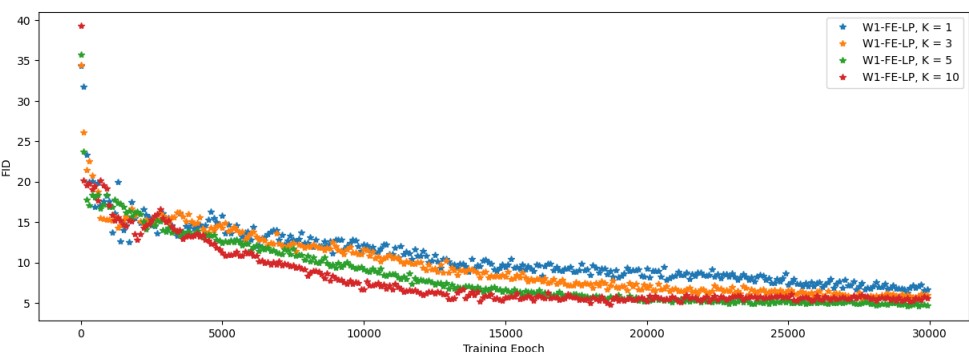

Figure 4: FID against training epoch for various W1-FE-LP models on generating CIFAR-10 images.

continuous time, whether $\mu^*(t)$ truly corresponds to a solution to ODE (3.6) is left unanswered. It is also unclear if $\mu^*(t)$ will ultimately converges to the data distribution $\mu_{\rm d}$, although it is intuitively expected by our "gradient descent" idea. To fill these gaps, one wishes to show that (i) there exists a (unique) solution $Y$ to ODE (3.6), (ii) the law of $Y_t$ coincides with $\mu^*$, i.e., $\mu^{Y_t} = \mu^*(t)$ for all $t \geq 0$, and (iii) $\mu^{Y_t}$ ultimately converges to $\mu_{\rm d}$, i.e., $W_1(\mu^{Y_t}, \mu_{\rm d}) \to 0$ as $t \to \infty$. The challenge here is twofold. First, as the coefficient $(\mu, x) \mapsto \nabla\varphi_\mu^{\mu_{\rm d}}(x)$ of (3.6) is not continuous in general, standard existence results for distribution-dependent (stochastic) differential equations (i.e., McKean-Vlasov equations) cannot be easily applied. Second, when analyzing the flow of measures $\{\mu^{Y_t}\}_{t \geq 0}$ in $\mathcal{P}_1(\mathbb{R}^d)$ through the continuity equation (or, Fokker-Planck equation) associated with (3.6), the standard theory in Ambrosio et al. (2008) does not provide concrete results or guidance, as it covers the case $\mathcal{P}_p(\mathbb{R}^d)$ for all $p > 1$, but excludes our present case $\mathcal{P}_1(\mathbb{R}^d)$.

Numerically, while we have shown that persistent training can markedly improve training results in several experiments, it is not without restrictions. Recall that SGD is performed $K \in \mathbb{N}$ consecutive times with the same minibatch $\{\zeta_i\}$ in Algorithm 1, with $\zeta_i = G_\theta(z_i) - \epsilon\nabla\varphi(G_\theta(z_i))$. There are two issues one has to confront. First, any inaccuracy in the estimation of the Kantorovich potential $\varphi$ will be amplified by persistent training. As a larger $K \in \mathbb{N}$ demands our algorithm to more closely fits the data points $\{\zeta_i\}$, even when $\{\zeta_i\}$ are of low quality due to the inaccuracy of $\varphi$, the issue of "garbage in, garbage out" will be exacerbated. Second, even if $\varphi$ is perfectly estimated, such that $\{\zeta_i\}$ are of high quality, an excessive $K \in \mathbb{N}$ will certainly lead to overfitting.

The first issue can be mitigated by better estimation of the Kantorovich potential $\varphi$. Indeed, the reason why the experiments in Section 5 are run using W1-FE-LP, but not W1-FE-GP, is that the former is known to estimate $\varphi$ more accurately than the latter (Petzka et al., 2018); recall the distinction between the two algorithms in the paragraph below Remark 4.1. As a simple demonstration, in Appendix B.1, we run the first experiment in Section 5 again using W1-FE-GP and compare the results with those under W1-FE-LP previously presented. It shows that raising persistency levels

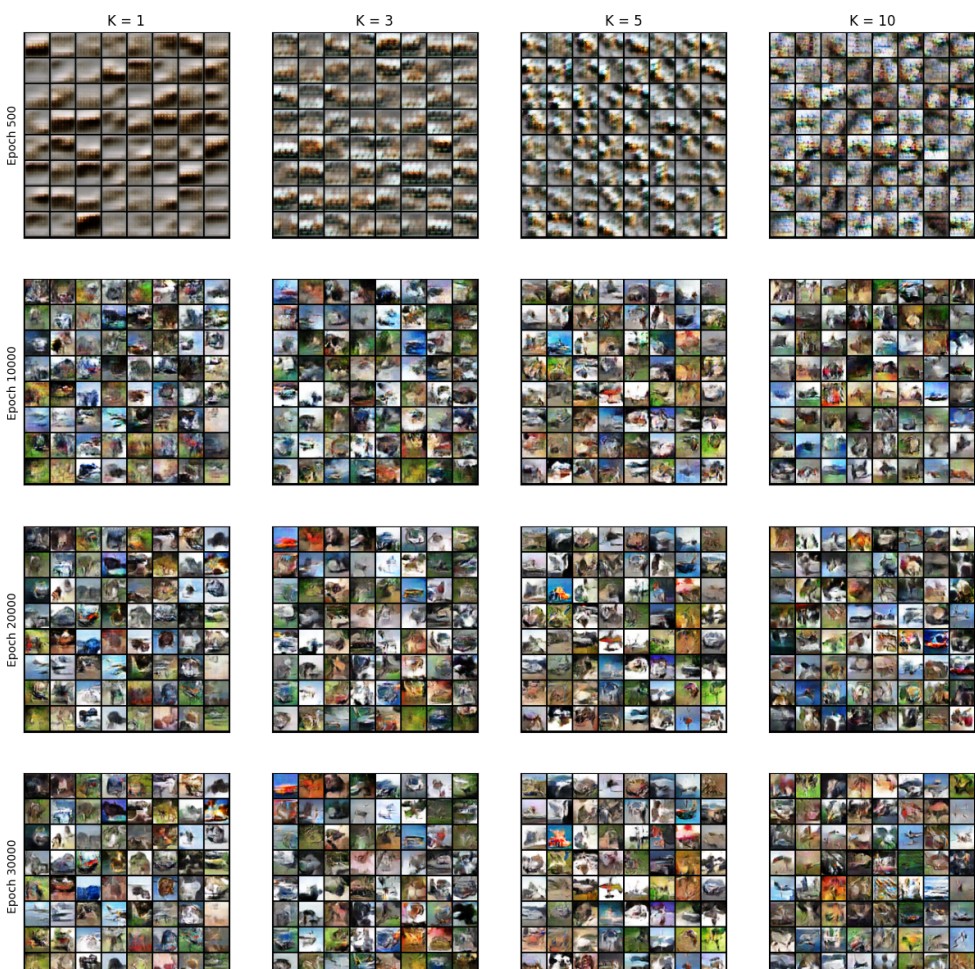

Figure 5: Uncurated samples from various W1-FE-LP models across training.

results in significantly more severe instability under W1-FE-GP than under W1-FE-LP. On the other hand, to mitigate the overfitting issue, we suggest finding a suitable persistency level through careful numerical investigation. For instance, the experiments in Section 5 all indicate a threshold of $K \in \mathbb{N}$ beyond which the performance starts to deteriorate (i.e., $K = 5$, $K = 3$, and $K = 5$ in the first, second, and third experiments, respectively). Taking $K \in \mathbb{N}$ to be at such a threshold (but not beyond it) can likely balance the benefits of persistent training against overfitting.

# 7 CONCLUSION

By performing "gradient descent" in the space $\mathcal{P}_1(\mathbb{R}^d)$, we introduce a distribution-dependent ODE for the purpose of generative modeling. A forward Euler discretization of the ODE converges to a curve of probability measures, suggesting that numerical implementation of the discretization is stable for small time steps. This inspires a class of new algorithms (called W1-FE) that naturally involves persistent training. If we (artificially) choose not to implement persistent training, our algorithms recover existing WGAN algorithms. By increasing the level of persistent training suitably (to better simulate the ODE), our algorithms outperform existing WGAN algorithms in numerical examples.

# 8 ACKNOWLEDGEMENTS

We thank Google for their freely available Google Colab, which we used for all simulations.

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

# A THEORETICAL RESULTS

## A.1 CONVEXITY OF $W_1(\cdot, \mu_d)$

**Proposition A.1.** *The function $J : \mathcal{P}_1(\mathbb{R}^d) \to \mathbb{R}$ in (3.3) is convex. That is, for any $\mu, \nu \in \mathcal{P}_1(\mathbb{R}^d)$ and $\lambda \in (0, 1)$, we have $J((1 - \lambda)\mu + \lambda\nu) \leq \lambda J(\mu) + (1 - \lambda)J(\nu)$.*

*Proof.* For any $\mu, \nu \in \mathcal{P}_1(\mathbb{R}^d)$ and $\lambda \in (0, 1)$, let $\varphi_\lambda$ denote a Kantorovich potential from $(1 - \lambda)\mu + \lambda\nu \in \mathcal{P}_1(\mathbb{R}^d)$ to $\mu_{\mathrm{d}}$. By (3.3), (2.2), and Definition 2.1,

$$J((1-\lambda)\mu + \lambda\nu) = W_1((1-\lambda)\mu + \lambda\nu, \mu_{\mathrm{d}}) = \int_{\mathbb{R}^d} \varphi_\lambda \, d((1-\lambda)\mu + \lambda\nu) - \int_{\mathbb{R}^d} \varphi_\lambda \, d\mu_d$$

$$= (1-\lambda) \int_{\mathbb{R}^d} \varphi_\lambda \, d(\mu - \mu_d) + \lambda \int_{\mathbb{R}^d} \varphi_\lambda \, d(\nu - \mu_d)$$

$$\leq (1-\lambda) \sup_{||\varphi||_{\mathrm{Lip}} \leq 1} \left\{ \int_{\mathbb{R}^d} \varphi \, d(\mu - \mu_d) \right\} + \lambda \sup_{||\varphi||_{\mathrm{Lip}} \leq 1} \left\{ \int_{\mathbb{R}^d} \varphi \, d(\nu - \mu_d) \right\}$$

$$= (1-\lambda)J(\mu) + \lambda J(\nu),$$

where the last equality follows again from (3.3) and (2.2). □

**Remark A.1.** *In most cases, the inequality in the proof above is strict, as it is in general unlikely that $\varphi_\lambda$ also attains both of the two suprema.*

## A.2 PROOF OF PROPOSITION 3.1

*Proof.* Fix $\mu, \nu \in \mathcal{P}_1(\mathbb{R}^d)$. For any $\epsilon \in (0, 1)$, note that $\mu + \epsilon(\nu - \mu) = (1 - \epsilon)\mu + \epsilon\nu$ remains in $\mathcal{P}_1(\mathbb{R}^d)$. By (3.3) and Definition 2.1,

$$J(\mu) = W_1(\mu, \mu_{\mathrm{d}}) = \int_{\mathbb{R}^d} \varphi_\mu^{\mu_d} \, d(\mu - \mu_{\mathrm{d}}), \tag{A.1}$$

$$J(\mu + \epsilon(\nu - \mu)) = W_1(\mu + \epsilon(\nu - \mu), \mu_{\mathrm{d}}) = \int_{\mathbb{R}^d} \varphi_{\mu+\epsilon(\nu-\mu)}^{\mu_d} \, d(\mu + \epsilon(\nu - \mu) - \mu_{\mathrm{d}}). \tag{A.2}$$

On the other hand, by (3.3), the duality formula (2.2), and the fact that $\varphi_\mu^{\mu_{\mathrm{d}}}, \varphi_{\mu+\epsilon(\nu-\mu)}^{\mu_{\mathrm{d}}} : \mathbb{R}^d \to \mathbb{R}$ are 1-Lipschitz functions, we obtain the inequalities

$$J(\mu) = W_1(\mu, \mu_{\mathrm{d}}) \geq \int_{\mathbb{R}^d} \varphi_{\mu+\epsilon(\nu-\mu)}^{\mu_{\mathrm{d}}} \, d(\mu - \mu_{\mathrm{d}}), \tag{A.3}$$

$$J(\mu + \epsilon(\nu - \mu)) = W_1(\mu + \epsilon(\nu - \mu), \mu_{\mathrm{d}}) \geq \int_{\mathbb{R}^d} \varphi_\mu^{\mu_{\mathrm{d}}} \, d(\mu + \epsilon(\nu - \mu) - \mu_{\mathrm{d}}). \tag{A.4}$$

It follows from (A.4) and (A.1) that

$$J(\mu + \epsilon(\nu - \mu)) - J(\mu) \geq \int_{\mathbb{R}^d} \varphi_\mu^{\mu_\mathrm{d}} \, d(\mu + \epsilon(\nu - \mu) - \mu_\mathrm{d}) - \int_{\mathbb{R}^d} \varphi_\mu^{\mu_\mathrm{d}} \, d(\mu - \mu_\mathrm{d})$$

$$= \epsilon \int_{\mathbb{R}^d} \varphi_\mu^{\mu_\mathrm{d}} \, d(\nu - \mu),$$

while (A.2) and (A.3) imply

$$J(\mu + \epsilon(\nu - \mu)) - J(\mu) \leq \int_{\mathbb{R}^d} \varphi_{\mu + \epsilon(\nu - \mu)}^{\mu_\mathrm{d}} \, d(\mu + \epsilon(\nu - \mu) - \mu_\mathrm{d}) - \int_{\mathbb{R}^d} \varphi_{\mu + \epsilon(\nu - \mu)}^{\mu_\mathrm{d}} \, d(\mu - \mu_d)$$

$$= \epsilon \int_{\mathbb{R}^d} \varphi_{\mu + \epsilon(\nu - \mu)}^{\mu_\mathrm{d}} \, d(\nu - \mu).$$

Putting the above two inequalities together, we see that

$$\int_{\mathbb{R}^d} \varphi_\mu^{\mu_\mathrm{d}} \, d(\nu - \mu) \leq \frac{J(\mu + \epsilon(\nu - \mu)) - J(\mu)}{\epsilon} \leq \int_{\mathbb{R}^d} \varphi_{\mu + \epsilon(\nu - \mu)}^{\mu_\mathrm{d}} \, d(\nu - \mu).$$

As $\epsilon \to 0^+$, since $\varphi_{\mu + \epsilon(\nu - \mu)}^{\mu_d}$ converges uniformly to $\varphi_\mu^{\mu_d}$ on compacts of $\mathbb{R}^d$ (Santambrogio, 2015, Theorem 1.52), the right-hand side above tends to $\int_{\mathbb{R}^d} \varphi_\mu^{\mu_d} \, d(\nu - \mu)$. This then implies

$$\lim_{\epsilon \to 0^+} \frac{J(\mu + \epsilon(\nu - \mu)) - J(\mu)}{\epsilon} = \int_{\mathbb{R}^d} \varphi_\mu^{\mu_\mathrm{d}} \, d(\nu - \mu),$$

i.e., $\varphi_\mu^{\mu_\mathrm{d}}$ is a linear functional derivative of $J$. $\qquad\square$

## A.3 A Refined Arzela-Ascoli Result

The following is a transcription of Ambrosio et al. (2008, Proposition 3.3.1) in our specific setting, where we consider the metric space $\mathcal{P}_1(\mathbb{R}^d)$ with the natural topology induced by the $W_1$ distance.

**Proposition A.2.** *Fix $T > 0$ and let $K \subseteq \mathcal{P}_1(\mathbb{R}^d)$ be compact in $\mathcal{P}_1(\mathbb{R}^d)$ under the topology induced by the $W_1$ distance. For any sequence $\{g_n\}_{n \in \mathbb{N}}$ of curves $g_n : [0, T] \to \mathcal{P}_1(\mathbb{R}^d)$ such that*

$$g_n(t) \in K, \quad \forall n \in \mathbb{N}, \ t \in [0, T], \tag{A.5}$$

$$\limsup_{n \to \infty} W_1(g_n(s), g_n(t)) \leq |s - t| \quad \forall s, t \in [0, T], \tag{A.6}$$

*there exist an increasing subsequence $k \to n(k)$ and a continuous $g : [0, T] \to \mathcal{P}_1(\mathbb{R}^d)$ such that*

$$W_1(g_{n(k)}(t), g(t)) \to 0 \quad \forall t \in [0, T]. \tag{A.7}$$

## A.4 Proof of Theorem 4.1

*Proof.* Let $(\Omega, \mathcal{F}, \mathbb{P})$ be the underlying probability space that supports all the random variables $\{Y_{n-1,\epsilon} : n \in \mathbb{N}, \epsilon > 0\}$, defined as in (4.1). Fix any $T > 0$. We will show that $\{\mu_\epsilon(t) : \epsilon > 0, t \in [0, T]\}$ fulfills (A.5) and (A.6). For any fixed $t \in [0, T]$, there exists $n \in \mathbb{N}$ such that $t \in [(n-1)\epsilon, n\epsilon)$ and $\mu_\epsilon(t) = \mu^{Y_{n-1,\epsilon}}$. By (4.1), the random variable $Y_{n-1,\epsilon}$ takes the form

$$Y_{n-1,\epsilon} = Y_0 - \epsilon \sum_{i=0}^{n-2} \nabla \varphi_\mu^{\mu_d^{Y_{i,\epsilon}}} (Y_{i,\epsilon}). \tag{A.8}$$

As $|\nabla \varphi_\mu^\nu(x)| \leq 1$ $\mathcal{L}^d$-a.e. on $\mathbb{R}^d$ for all $\mu, \nu \in \mathcal{P}_1(\mathbb{R}^d)$ (Remark 2.1), this implies

$$|Y_{n-1,\epsilon}| \leq |Y_0| + (n-1)\epsilon \leq |Y_0| + t \leq |Y_0| + T \quad \text{a.s.}, \tag{A.9}$$

where the second inequality is due to $t \in [(n-1)\epsilon, n\epsilon)$. It follows that for all $t \in [0, T]$,

$$\sup_{\epsilon > 0} \int_{\mathbb{R}^d} |y| \, d\mu_\epsilon(t) = \sup_{\epsilon > 0} \int_{\mathbb{R}^d} |y| \, d\mu^{Y_{n-1,\epsilon}} = \sup_{\epsilon > 0} \mathbb{E}^\mathbb{P}[|Y_{n-1,\epsilon}|] \leq \mathbb{E}^\mathbb{P}[|Y_0|] + T < \infty, \tag{A.10}$$

By (A.10) and the fact that the function $\phi(y) := |y|$, $y \in \mathbb{R}^d$, has compact sublevels (i.e., the set $\{y : |y| \le c\}$ is compact in $\mathbb{R}^d$ for any $c \ge 0$), Ambrosio et al. (2008, Remark 5.1.5) asserts that the collection of measures $\{\mu_\epsilon(t) : \epsilon > 0, t \in [0, T]\}$ is tight (i.e., precompact under the topology of weak convergence). To further prove that this collection of measures is precompact in $\mathcal{P}_1(\mathbb{R}^d)$, it suffices to show that the measures have uniformly integrable first moments, in view of Ambrosio et al. (2008, Proposition 7.1.5). That is, we need to show that

$$\lim_{k \to \infty} \sup_{\epsilon > 0, t \in [0,T]} \int_{\mathbb{R}^d \setminus B_k(0)} |y| \, d\mu_\epsilon(t) = 0,$$

where $B_k(0)$ denotes the open ball centered at $0 \in \mathbb{R}^d$ with radius $k > 0$. For any fixed $t \in [0, T]$, by the same arguments above (A.8),

$$\int_{\mathbb{R}^d \setminus B_k(0)} |y| \, d\mu_\epsilon(t) = \mathbb{E}^{\mathbb{P}} \left[ |Y_{n-1,\epsilon}| \, \mathbb{I}_{\mathbb{R}^d \setminus B_k(0)}(Y_{n-1,\epsilon}) \right]$$

$$= \mathbb{E}^{\mathbb{P}} \left[ |Y_{n-1,\epsilon}(\omega)| \, \mathbb{I}_{\{|Y_{n-1,\epsilon}(\omega)| \ge k\}}(\omega) \right]$$

$$\le \mathbb{E}^{\mathbb{P}} \left[ |Y_0(\omega) + T| \, \mathbb{I}_{\{|Y_0(\omega)+T| \ge k\}}(\omega) \right]$$

where $\mathbb{I}$ denotes an indicator function and the inequality follows from (A.9). Hence,

$$\sup_{\epsilon > 0, t \in [0,T]} \int_{\mathbb{R}^d \setminus B_k(0)} |y| \, d\mu_\epsilon(t) \le \mathbb{E}^{\mathbb{P}} \left[ |Y_0(\omega) + T| \, \mathbb{I}_{\{|Y_0(\omega)+T| \ge k\}}(\omega) \right] \to 0 \quad \text{as } k \to \infty,$$

where the convergence follows from $Y_0 \in L^1(\mathbb{P})$, thanks to $\mu^{Y_0} = \mu_0 \in \mathcal{P}_1(\mathbb{R}^d)$. We therefore conclude that $\{\mu_\epsilon(t) : \epsilon > 0, t \in [0, T]\}$ is precompact in $\mathcal{P}_1(\mathbb{R}^d)$ and thus fulfills (A.5).

Next, consider any $s, t \in [0, T]$ with $s \ne t$. Without loss of generality, assume $s < t$. For any fixed $\epsilon > 0$, there exist $j, k \in \mathbb{N}$ with $j \le k$ such that

$$(j-1)\epsilon \le s < j\epsilon \quad \text{and} \quad \mu_\epsilon(s) = \mu^{Y_{j-1,\epsilon}}; \quad (k-1)\epsilon \le t < k\epsilon \quad \text{and} \quad \mu_\epsilon(t) = \mu^{Y_{k-1,\epsilon}}. \quad \text{(A.11)}$$

By (4.1), we have

$$Y_{k-1,\epsilon} = Y_{j-1,\epsilon} - \epsilon \sum_{i=1}^{k-j} \nabla \varphi_{\mu}^{\mu_d}{}_{Y_{i-1,\epsilon}}(Y_{i,\epsilon}). \quad \text{(A.12)}$$

It follows that

$$W_1(\mu_\epsilon(s), \mu_\epsilon(t)) = W_1(\mu^{Y_{j-1,\epsilon}}, \mu^{Y_{k-1,\epsilon}})$$

$$\le \mathbb{E}^{\mathbb{P}}[|Y_{k-1,\epsilon} - Y_{j-1,\epsilon}|] = \mathbb{E}^{\mathbb{P}} \left[ \left\| \epsilon \sum_{i=1}^{k-j} \nabla \varphi_{\mu}^{\mu_d}{}_{Y_{n-1,\epsilon}}(Y_{i,\epsilon}) \right\| \right] \quad \text{(A.13)}$$

$$\le \epsilon(k-j) < |t - s| + \epsilon,$$

where the second inequality follows again from $|\nabla \varphi_\mu^\nu(x)| \le 1$ $\mathcal{L}^d$-a.e. on $\mathbb{R}^d$ for all $\mu, \nu \in \mathcal{P}_1(\mathbb{R}^d)$ (Remark 2.1) and the third inequality is due to (A.11). This immediately yields

$$\limsup_{\epsilon \to 0} W_1(\mu_\epsilon(s), \mu_\epsilon(t)) \le |s - t|, \quad \forall s, t \in [0, T], \quad \text{(A.14)}$$

i.e., $\{\mu_\epsilon(t) : \epsilon > 0, t \in [0, T]\}$ satisfies (A.6).

Now, we can apply Proposition A.2 to obtain a subsequence $\{\epsilon_k\}$ and a continuous curve $\mu_T^*(t) : [0, T] \to \mathcal{P}_1(\mathbb{R}^d)$ such that $W_1(\mu_{\epsilon_k}(t), \mu_T^*(t)) \to 0$ for all $t \in [0, T]$. By a diagonal argument, we can construct a continuous $\mu^* : [0, \infty) \to \mathcal{P}_1(\mathbb{R}^d)$ such that $W_1(\mu_{\epsilon_k}(t), \mu^*(t)) \to 0$ for all $t \ge 0$, possibly along a further subsequence. $\qquad \square$

## B   MORE EXPERIMENTAL RESULTS

### B.1   PERSISTENCY ON W1-FE-GP

We run the first experiment in Section 5 again using W1-FE-GP. The results, along with those under W1-FE-LP in the main text, are shown in Figure 6. As we can see, raising persistency levels results in significantly more severe instability under W1-FE-GP than under W1-FE-LP.

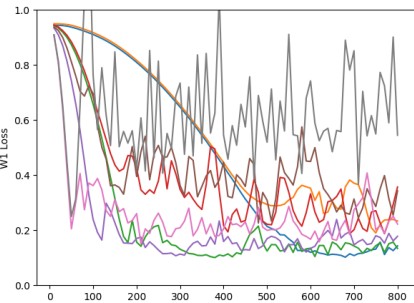 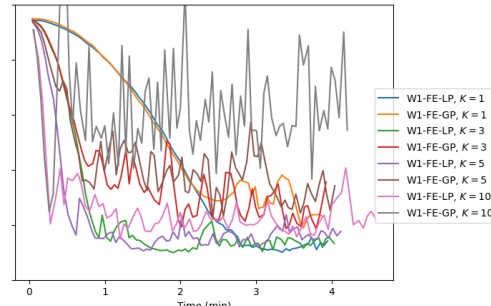

Figure 6: $W_1$ loss of W1-FE-GP and W1-FE-LP with persistency levels $K = 1, 3, 5, 10$ against training epoch (left) and wallclock time (right), respectively.

## C  USING THE CODE

We built off of the software package developed for use in Leygonie et al. (2019). While we made substantial changes to the package for our own purposes, we do acknowledge that the package built by Leygonie et al. (2019) made it substantially easier for us to implement our algorithm. The usage is almost identical to the original package's usage.

We recommend storing the code as either a zipped file or pulling directly from the GitHub repository. We also recommend using a Google Colab notebook as the virtual environment. Once the software package is loaded in the appropriate folder, one may reproduce the low dimensional experiments by running `main.py` inside `exp_2d`. The high dimensional experiments may be reproduced by running `main.py` inside `exp_da`.

If one uses Google Colab to run the experiments, then the default environment provided by the Google Colab Jupyter notebook in addition to the package Python Optimal Transport (POT) is required to run the software. To reproduce the plots, one needs the package `tensorboard`.

