# OpenReview forum: "A Differential Equation Approach for Wasserstein GANs and Beyond"
_ICLR.cc/2025/Conference — Submitted to ICLR 2025_

### Official Review · Reviewer_y5eR · 2024-10-31

**Soundness:** 2
**Presentation:** 3
**Contribution:** 2
**Rating:** 5
**Confidence:** 4

**Summary:**

This paper provides a new perspective on WGAN from the view of ODE associated with the gradient of Wasserstein derivative, demonstrating that persistent training on a generator can improve WGAN training.

**Strengths:**

1. This paper provides a clear and novel explanation of WGAN's training from the perspective of the gradient flow of Wasserstein distance.

2. Both theoretical explanations and experiments on toy and MNIST datasets demonstrate the effectiveness of persistent training, a common trick for training WGAN.

**Weaknesses:**

1. The main contributions in this paper, i.e. discretization and persistent training, are common tricks for training WGANs[1,2], which are not novel enough in practical implementation. For example, as is shown in the proof of Proposition 4.1, persistent training seems equal to just increasing the generator's iterations in the original WGAN's training. Thus please clarify how discretization and persistent training differ from the above existing methods.

2. Obtaining Kantorovich potential is a challenging and important step in ODE-based WGAN's training, but in this paper, it's still the same as the original WGAN, leaving some problems for persistent training as discussed in Remark 4.2, harming the consistency between theory and practice.

3. Experimental validation is not comprehensive enough, the used datasets are too small scale, like in WGAN and WGAN-GP,  more common and large-scale baseline datasets should also be verified. For example, how does the proposed method perform on CIFAR-10 or CelebA compared to baseline WGAN methods in terms of FID, IS, and convergence speed?

4. Some mathematical explanations and notations should be modified. For example, there is no definition of $m$ in Eq.(2.4). Besides, more introduction of $\nabla \frac{\delta J}{\delta m}$ are suggested to be added, since $\nabla \frac{\delta J}{\delta m}$ is very important in the whole method, how to understand this gradient and how to get Eq(3.2) from Eq(3.1) should be added in the main paper.

[1] Variational Wasserstein gradient flow. ICML 2022.

[2] Scalable Wasserstein Gradient Flow for Generative Modeling through Unbalanced Optimal Transport. ICML 2024.

**Questions:**

1. please clarify how persistent training differs from simply increasing generator iterations.

2. As discussed in the third point of the above weaknesses, more common large-scale datasets should also be verified.

---

> ### Author Response · Authors · 2024-11-27
>
> We sincerely thank the reviewer for providing valuable constructive feedback. We respond to the comments point by point below.
>
> **Weaknesses (W):**
>
> 1.	Let us separate our response into three parts.
>
>    - Indeed, there are many recent papers that feature “Wasserstein gradient flows” and consider discretization and persistent training, including the two mentioned by the reviewer. But a closer examination reveals that they don’t actually have a direct connection to Wasserstein GAN (WGAN). The common theme of these papers is to minimize a loss function using the well-developed gradient flows under the Wasserstein-2 distance. In contrast to this, WGAN by construction minimizes the Wasserstein-1 distance and cannot be easily analyzed by the standard Wasserstein-2 framework. A main contribution of our paper is to provide a rigorous study on Wasserstein-1 gradient flows, which are much less understood in the literature. This allows us to build up a theoretical framework that not only recovers WGAN but suggests improvements to it (which leads to our algorithm W1-FE).
>
>    -  The second last paragraph in the introduction now presents the explanations above, to distinguish our paper from others that also feature “Wasserstein gradient flows.” In addition, the second, third, and fourth paragraphs in the introduction discuss in detail the major theoretical challenges we face in studying Wasserstein-1 gradient flows, as opposed to using the full-fledged theory of Wasserstein-2 gradient flows directly  (as is done by other papers), and how we overcome these challenges in this paper.
>
>    - On the other hand, we stress that our algorithm W1-FE is fundamentally different from WGAN. While Proposition 4.1 shows that W1-FE with persistency level $K=1$ (i.e., no persistent training at all) reduces to WGAN, this is in fact the only case where they coincide. In the newly revised Remark 4.2, we explain in detail that even if we also enforce persistent training in WGAN, W1-FE and WGAN will differ from each other starting from $K=2$. This is because the equality in the proof of Proposition 4.1 holds under $K=1$, but fails in general for any $K>1$. It is actually not surprising to see this general distinction between W1-FE and WGAN. As they are built from fundamentally different ideas for generative modeling (see the discussion below Proposition 4.1), how generators are updated in these two algorithms are quite different and only coincide in the simplest case $K=1$.
>
> 2.	Remark 4.2 has been substantially rewritten to possibly avoid any confusion. As mentioned above, what we hope to convey in Remark 4.2 is that even when persistent training is incorporated into WGAN, WGAN is still quite different from our algorithm W1-FE (except the simplest case $K=1$, i.e., no persistent training). In particular, W1-FE is not the same as adding persistent training to the generator update in WGAN.
>
> 3.	In Section 5, we consider a new experiment where we train our algorithm on the CIFAR-10 dataset. The results, presented in Figures 4 and 5, particularly show that W1-FE with $K>1$ outperforms W1-FE with $K=1$ (i.e., WGAN) in terms of both the eventual FID achieved and convergence speed.
>
> 4.	Let us separate our response into three parts.
>
>    - The “$m$” that appears in Definition 3.1 is only a common notation to remind us that the variable in discussion is a probability measure. There is no formal mathematical definition of it. This is now mentioned below Definition 3.1.
>
>    - To better explain the meaning of $\nabla\frac{\delta J}{\delta m}$, we now clearly state a gradient-type property satisfied by $\nabla\frac{\delta J}{\delta m}$ (i.e., the first equation on p. 4). It shows that when points $y\in\mathbb{R}^d$ are moved by a vector field $\xi:\mathbb{R}^d\to\mathbb{R}^d$, $\nabla\frac{\delta J}{\delta m}(\mu,y)$  serves to specify how moving along $\xi(y)$ instantaneously changes the value of $J$.
>
>    - To explain how the ODE (3.5) is obtained from our problem (3.1), we first recall the classical gradient-descent ODE in (3.2) for minimizing a convex function $f:\mathbb{R}^d\to\mathbb{R}$. The essence of this classical ODE is to move along the negative gradient of the convex function $f$. For our problem (3.1), which is the minimization of the convex function $J:\mathcal{P}_1(\mathbb{R}^d)\to\mathbb{R}$, we hope to derive a gradient-descent ODE similar to (3.2). Once we take $\nabla\frac{\delta J}{\delta m}$ to be the gradient of $J$ (as argued in the previous paragraph), the idea of moving along the negative gradient of $J$ then yields the ODE (3.5). In the line above (3.5), we particularly mention that the derivation of (3.5) is in analogy to the classical ODE (3.2).
>
> **Questions (Q):**
>
> 1.	Please see our explanations above in the last part of W1 and also in W2.
>
> 2.	We have added a new experiment where our algorithm is trained on the CIFAR-10 dataset. Please see W3 above.

---

> > ### Comment · Reviewer_y5eR · 2024-11-28
> >
> > Thanks for the author's response and clear explanation of the difference between WGAN and WGAN-FE. I recognize the theoretical contribution of this paper for building a connection between WGAN and W1 gradient flow. However, to meet ICLR's standard, I'm afraid some aspects— experimental or theoretical—still require further improvement or clarification. Below, I outline my concerns:
> >
> > 1.  This paper trains a generator using the gradient flow of W1 distance. Similarly, other works trained generators using the Wasserstein gradient flow of KL divergence or F-divergence [1, 2]. In [1], the training approach appears very similar to the method presented in this paper (see Eq.(17) in [1]). This raises the question: what distinguishes using the gradient flow of the Wasserstein distance from that of F-divergence? Furthermore, does employing the Wasserstein distance offer any specific advantages? In [2], for instance, using the Langevin dynamic can alleviate mode collapse, but it appears that the proposed method does not exhibit this property. Understanding why using the Wasserstein distance rather than other divergences is important and meaningful for training generative models. However, to the best of my knowledge, this question has not been sufficiently addressed so far. Can the authors give some insights into this question? Or at the very least, some experimental results are necessary to provide a comparison between these approaches.
> >
> > 2. The experiments are still too simple. Given the rapid development of generative models in recent years, the experimental setup and results in this paper are not sufficient to meet the current standards of research in this field. For instance, this paper provides only qualitative results on CIFAR-10 but does not include quantitative evaluations. Are there measurable advantages over WGAN-GP or other GAN-based methods? Additionally, Figures 4 and 5 focus solely on ablation studies and do not include comparisons with other types of models, which limits the scope of the analysis.
> >
> > 3. Although the optimization of the generator in this paper is theoretically different from that in WGAN, what happens if the number of generator training iterations is increased within each mini-max optimization step? Would the proposed method still offer any advantages in such a scenario?
> >
> > Based on the above concerns, in the current stage, I tend to maintain my score.
> >
> > [1] MonoFlow: Rethinking Divergence GANs via the Perspective of Differential Equations. ICML 2023
> > [2] Cooperative Learning of Energy-Based Model and Latent Variable Model via MCMC Teaching. AAAI 2018

---

> > > ### Author Response · Authors · 2024-12-04
> > >
> > > We thank the reviewer again for the feedback, and we respond to the comments point by point below.
> > >
> > > 1.	We agree with the reviewer that it is important to compare gradient flows of Wasserstein-1 distance and those of $f$-divergence. A logical implementation of this should contain three steps.
> > >
> > >    **Step 1:** Show that gradient flows of Wasserstein-1 distance are theoretically well-defined and design a corresponding algorithm (i.e., W1-FE in our paper).
> > >
> > >    **Step 2:** Show that W1-FE, as an extension of WGAN, outperforms WGAN.
> > >
> > >    **Step 3:** Compare W1-FE with algorithms induced by gradient flows of $f$-divergence.
> > >
> > >    - Note that Step 2 is important: if W1-FE actually underperforms WGAN, one should instead compare WGAN with gradient flows of $f$-divergence in Step 3. After all, logically what’s the point of comparing an inferior algorithm with other kinds of algorithms? One hopes to compare a superior algorithm with other kinds of algorithms to find an even better one.
> > >
> > >    - Our paper precisely focuses on Steps 1and 2, which already require nontrivial theoretical development and numerical analysis. Including also Step 3 will likely make our paper too lengthy and less focused, which need not meet the usual ICLR standard. As a result, in terms of presentation, it seems reasonable to report our findings under Steps 1 and 2 in the present paper and carry out Step 3 comprehensively in a separate follow-up paper.
> > >
> > >
> > >    - We can provide some insights into why gradient flows of Wasserstein-1 distance could outperform those of $f$-divergence. Wasserstein GAN (WGAN) [1] and $f$-divergence GAN ($f$-GAN) [2] are two well-known extensions of the original GAN framework. While they were proposed roughly at the same time (in 2017 and 2016) and have both been popular since then, the popularity of WGAN is particularly phenomenal. For instance, [1] has been cited 17,600 times and some modified versions of WGAN are also highly cited, such as [3] (12,200 times). Such popularity of WGAN stems from its enhanced stability, which results from its ability to alleviate mode collapse and thus facilitate convergence [1] [4]. By contrast, [2] has been cited only 1,980 times and many versions of $f$-GAN are known to suffer mode collapse severely. In view of this, it is not unreasonable to conjecture that gradient flows of Wasserstein-1 distance (which build upon and improve WGAN) can outperform gradient flows of $f$-divergence (which build upon and improve $f$-GAN).
> > >
> > > 2.	We are somewhat caught off guard by this comment.
> > >
> > >    - First, in weakness #3, the reviewer only suggested that we use a more common and large-scale dataset, such as CIFAR-10. And we did exactly that in our revision. If the newly-raised questions had been communicated in the first place, we would have had more time to properly address them in our revision.
> > >
> > >    - Second, we don’t exactly understand the comment “…provides only qualitative results on CIFAR-10 but does not include quantitative evaluations.” In fact, we quantitatively evaluate algorithms’ performance on CIFAR-10 by computing the evolution of Fréchet inception distance (FID) in Figure 4, alongside qualitative results in Figure 5.
> > >    - Third, as mentioned in the paper, we use WGAN-LP instead of WGAN-GP because the former is known to outperform the latter in the literature.
> > >    - Fourth, as mentioned under 1, our paper focuses on Steps 1 and 2. For the purpose of Step 2, ablation studies are enough and there is no need to have comparisons with many other types of models. Such comparisons belong to Step 3 and, as argued above in 1, it could be reasonable to relegate Step 3 to a separate follow-up paper.
> > >
> > > 3.	Yes, our proposed method still offers advantages, based on an additional training experiment on CIFAR-10.
> > > Specifically, we allowed persistent training in the generator update of WGAN and trained this modified WGAN on CIFAR-10. The results are significantly worse than those under our proposed method W1-FE.  If there is a chance, we can add this additional experiment in the final version of our paper.
> > >
> > >
> > > [1] Wasserstein generative adversarial networks. ICML 2017.
> > >
> > > [2] $f$-GAN: Training Generative Neural Samplers using Variational Divergence Minimization. NIPS 2016.
> > >
> > > [3] Improved training of Wasserstein GANs. NIPS 2017.
> > >
> > > [4] Towards principled methods for training generative adversarial networks. ICLR 2017.

---

### Official Review · Reviewer_GQDK · 2024-11-04

**Soundness:** 3
**Presentation:** 3
**Contribution:** 3
**Rating:** 6
**Confidence:** 2

**Summary:**

This work analyzes Wasserstein GAN training through the lens of gradient flow dynamics derived from the optimal transport map between initial and data distributions. Authors demonstrate in Theorem 4.1 that by applying a sequence of updates to the generator distribution according to the gradient of the W1 witness function (Kantorovich potential), the resulting distribution in the limit matches the optimal one.
This framework motivates the use of *persistent training* of the generator, where at each substep of training the discriminator and generator noise $z$ is frozen while the generator $G_\theta$ is updated for K steps. Lastly, the authors use a few simplified experimental settings  (e.g. 2D mixture of Gaussians) to demonstrate how persistent training can improve the rate of training convergence.

**Strengths:**

* Utilizes tools from optimal transport for analyzing WGANs, providing a generalized WGAN training framework.
* Theory provides good insight into generator training hyperparameters, which is corroborated by experiments.

**Weaknesses:**

* It seems to me that most of the uncertainty about how well WGAN training follows idealized gradient flow dynamics lies with the discriminator. Can you still arrive at a similar conclusion to thoerem 4.1 when the distance between approximated + true potential function is bounded?
* I'd like to see more discussion in the introduction about key differences between contributions of this work and the W2-FE paper.

Minor Notes:

* Eq 2.2 \mu_t, \mu^d not defined initially
* both \varphi and \phi used on line 087
* Eq. 2.3 variable i is not defined

**Questions:**

* What was the number of steps used when training the discriminator in figure 1? Does your theory suggest when it is more or less important to have accurate $\varphi$ approximations during training?

---

> ### Author Response · Authors · 2024-11-27
>
> We sincerely thank the reviewer for providing valuable constructive feedback. We respond to the comments point by point below.
>
> **Weaknesses (W):**
>
> 1.	The convergence result (Theorem 4.1) actually holds quite generally, even when we replace the Kantorovich potential function $\varphi$ in the discretization (4.1) by another 1-Lipschitz function. Indeed, as the proof of Theorem 4.1 relies on only the fact that $\varphi$ is 1-Lipschitz, instead of the specific form of $\varphi$, the same convergence result still holds when $\varphi$ is replaced by another 1-Lipschitz function. This suggests that our discretization scheme is robust in the following sense: in actual computation, as long as the estimated $\varphi$ is 1-Lipschitz (which is facilitated by the discriminator’s regularization in Gulrajani et al. (2017) and Petzka et al. (2018)), the scheme remains stable for small time steps and there is a well-defined limit. All the explanations above are now presented in the newly added Remark 4.1.
>
> 2.	The introduction has been substantially rewritten to highlight the key differences between this work and the W2-FE paper. First, as we mention the second paragraph, it is tempting to suspect that generative modeling by minimizing Wasserstein-1 loss (our paper) can be achieved by slightly modifying the W2-FE paper, which minimizes Wasserstein-2 loss. Yet, there are two major theoretical hurtles unique to the Wasserstein-1 case (which are detailed in the third paragraph). First, it is not even clear how “gradient” should be defined under the Wasserstein-1 distance, as subdifferential calculus for the space of probability measures is well-developed under the Wasserstein-$p$ distance for any $p>1$ but breaks down exactly for $p=1$. Second, when showing that a discretization of the gradient-flow ODE under the Wasserstein-2 distance converges, the W2-FE paper crucially relies on an interpolation result from optimal transport, which holds under the Wasserstein-$p$ distance for any $p>1$, excluding exactly $p=1$. The fourth paragraph explains how we overcome the above two hurdles. First, we observe that a general gradient notion can be defined, independently of subdifferential calculus, by using linear functional derivatives from the mean field game literature. This allows us to precisely formulate the gradient-flow ODE under the Wasserstein-1 distance and devise a corresponding discretization. Next, without relying on any interpolation result from optimal transport, we prove the convergence of the ODE discretization using a refined Arzela-Ascoli argument. This argument can be applied because the ODE coefficient is uniformly bounded (which is unique to the $p=1$ case), allowing us to prove appropriate compactness and equicontinuity of the flow of measures induced by the discretization.
>
> 3.	The minor notes on notation have all been addressed. Please note that Section 2 has been made much more concise and focused than before and every equation has been examined closely to avoid possible confusion.
>
>
> **Questions (Q):**
>
> 1.	In Figure 1, the number of steps used when training the discriminator is 10. This is now mentioned in the second paragraph of Section 5.
>
> 2.	Our analysis suggests that it is more important to have accurate approximations of $\varphi$ (i.e., the Kantorovich potential) when the persistency level $K\in\mathbb{N}$ is larger. As explained in detail in the last two paragraphs of Section 6, any inaccuracy in $\varphi$ will be amplified by persistent training, which in turn exacerbates the “garbage in, garbage out” issue. Hence, when persistent training is intense (i.e., $K\in\mathbb{N}$ is large in our algorithm) and one enjoys the resulting accelerated convergence, accurate estimates of $\varphi$ are more pressingly needed to avoid “garbage in, garbage out.”

---

### Official Review · Reviewer_BHdK · 2024-11-04

**Soundness:** 2
**Presentation:** 1
**Contribution:** 2
**Rating:** 6
**Confidence:** 3

**Summary:**

This paper presents a new variant of Wasserstein generative adversarial networks (WGANs) based on a distribution-dependent ordinary differential equation (ODE). It introduces a method called W1 Forward Euler (W1-FE), which includes persistent training to improve efficiency. When persistent training is not utilized, the approach reverts to standard WGAN algorithms. By appropriately increasing the level of persistent training, the model's performance is enhanced.

**Strengths:**

This paper presents a novel framework for Wasserstein generative adversarial networks (WGANs) based on distribution-dependent ordinary differential equations (ODEs). It utilizes persistent training to enhance the training process of WGANs and provides a solid mathematical foundation for this approach. Additionally, the paper includes numerical results that demonstrate the effectiveness of persistent training at various levels.

**Weaknesses:**

The experiments are insufficient; the paper should include additional datasets or real-world applications. It is better to present empirical results using well-known standard datasets.

The mathematical framework needs further clarification. For instance, the inequality at the top of page 4 should be explained more thoroughly.

The section on "MATHEMATICAL PRELIMINARIES" could benefit from more references. Please include citations for definitions and related concepts.

The contributions seem to lack originality, as mentioned in paragraph 2 of the Introduction. This section suggests that the differential equation approach outlined in the paper is already well-established in the existing GANs literature. Therefore, the question to the author is: Is the approach a variant of Generative Modeling using Wasserstein-1 loss, achieved by minimizing Wasserstein-2 loss (Huang and Malik, 2024)? If the novelty is something else, please clarify this in the Introduction. See references from the papers itself

Y.-J. Huang and Z. Malik. Generative modeling by minimizing the wasserstein-2 loss, 2024. URL
https://arxiv.org/abs/2406.13619.

Y.-J. Huang and Y. Zhang. GANs as gradient flows that converge. Journal of Machine Learning
Research, 24(217):1–40, 2023. URL http://jmlr.org/papers/v24/22-0583.html.

**Questions:**

I have a few questions and suggestions as follows.

1.	For completeness, please include a proof for the statement, “$\mu \rightarrow W_1(\mu,\mu_d)$ is strictly convex on $P_1(X)$.”

2.	In Proposition 3.1, it would be helpful to explain the first inequality (at the top of page 4).

3.	The manuscript mentions the use of persistent training, which is known to have certain limitations. Could you elaborate on how these limitations are addressed or mitigated in your training approach?

4.	The manuscript's writing could benefit from some improvements, particularly in conveying the contributions of the proposed work more clearly and concisely.

5.	Please consider including some recent, relevant citations to enhance the manuscript's context within the field.

6.	The numerical experiments are limited in scope; please include more real-world datasets (such as CIFAR-10 or CIFAR-100) in the empirical results.

---

> ### Author Response · Authors · 2024-11-27
>
> We sincerely thank the reviewer for providing valuable constructive feedback. We respond to the comments point by point below.
>
> **Weaknesses (W):**
>
> A new training experiment on the CIFAR-10 dataset is now included (see Q6 below).
>
> We went through every mathematical derivation carefully and made changes or added explanations if necessary to enhance clarity. For the specific inequality the reviewer mentioned, please see Q2 below.
>
> The section on "Mathematical Preliminaries" (i.e., Section 2) has been made much more concise and focused than before. Every concept, definition, and stated result is now supported by clear citations.
>
> The introduction has been substantially rewritten to highlight (i) the theoretical challenges unique to the present case of Wasserstein-1 loss and (ii) how we made new theoretical progress to overcome these challenges. Please see Q4 below for details.
>
>
>
> **Questions (Q):**
>
> 1.	In Appendix A.1, we first prove that $\mu\mapsto W_1(\mu,\mu_d)$ is convex in $\mathcal{P}_1(\mathbb{R}^d)$ (Proposition A.1) and then explain from the proof arguments that in most cases it can actually be “strictly” convex (Remark A.1).
>
> 2.	The proof of Proposition 3.1 has been moved to Appendix A.2 and the inequality in question is now (A.3). We explain in detail above (A.3) that this inequality results from the definition of $J$ in (3.3), the duality formula of the $W_1$ distance in (2.2), and the fact that the involved Kantorovich potential $\varphi$ is 1-Lipschitz.
>
> 3.	In the last two paragraphs of Section 6, we first explain in detail the two common limitations of persistent training (i.e., the exacerbation of “garbage in, garbage out” and overfitting) and discuss potential methods to mitigating them.  Specifically, we suggest (i) using a better estimation method for Kantorovich potential in our algorithm and (ii) carrying out numerical investigation to find a persistency level that best balances the benefits of persistent training against its limitations. These two methods are already incorporated into our numerical experiments.
>
> 4.	We have substantially rewritten the introduction to better convey the contributions of our proposed work. First, we mention that it is tempting to suspect that generative modeling using Wasserstein-1 loss can be achieved by slightly modifying the previous work that minimizes Wasserstein-2 loss (the second paragraph). Yet, there are two major theoretical hurtles unique to the Wasserstein-1 case (detailed in the third paragraph). First, it is not even clear how “gradient” should be defined under the Wasserstein-1 distance, as subdifferential calculus for the space of probability measures is well-developed under the Wasserstein-$p$ distance for any $p>1$ but breaks down exactly for $p=1$. Second, when showing that a discretization of the gradient-flow ODE under the Wasserstein-2 distance converges, Huang and Malik (2024) crucially rely on an interpolation result from optimal transport, which again holds under the Wasserstein-$p$ distance for any $p>1$, excluding exactly $p=1$. The fourth paragraph explains how we overcome the above two hurdles. First, we observe that a general gradient notion can be defined, independently of subdifferential calculus, by using linear functional derivatives from the mean field game literature. This allows us to precisely formulate the gradient-flow ODE under the Wasserstein-1 distance and devise a corresponding discretization. Next, without relying on any interpolation result from optimal transport, we prove the convergence of the ODE discretization using a refined Arzela-Ascoli argument. This argument can be applied because the ODE coefficient is uniformly bounded (which is unique to the $p=1$ case), allowing us to prove appropriate compactness and equicontinuity of the flow of measures induced by the discretization.
>
> 5.	The second last paragraph in the introduction now cites a list of recent, relevant papers that also feature “Wasserstein gradient flows.” We point out that all these studies leverage on the well-developed gradient flow theory under the Wasserstein-2 distance. Our study is distinct from theirs, as we focus on Wasserstain-1 gradient flows, which are much less understood but necessary for making a connection to WGAN (Recall that WGAN by construction minimizes the Wasserstein-1 distance).
>
> 6.	We add to Section 5 a new experiment where we train our algorithm on the CIFAR-10 dataset. The results are presented in Figures 4 and 5.

---

### Author Response · Authors · 2024-11-28
**Revision Submitted**

We would like to thank all the reviewers for your valuable comments from various angles. We have carefully revised our paper according to every single comment you provided and have just submitted the revived version of our paper. While each one of you can see our detailed response to your comments right below your review, we would like to point out here two major changes made to our paper.

1. A new training experiment on the CIFAR-10 dataset is added.
2. In the introduction, we now explain in detail our contributions relative to other recent studies that also feature "Wasserstein gradient flows." In short, while almost all other papers rely on the well-developed gradient flow theory under the Wasserstein-2 distance, we focus on Wasserstein-1 gradient flows, which are much less understood and necessary for making a connection to Wasserstein GAN.


Sincerely,

The authors

---

### Meta-Review · Area_Chair_fXmA · 2024-12-17

**Metareview:**

This paper is concerned with the Wasserstein generative adversarial network (WGAN). It introduces a notation of gradient flow associated with WGAN leveraging the linear functional derivative. Based on it, the authors propose an algorithm to train WGAN. The major criticism is on the contribution. The proposed method resembles existing methods that utilizes gradient flow associated with Wasserstein-2 metric or f-divergence. The only difference is on the computation of the potential function (discriminator). In addition, the experiments are insufficient to demonstrate the advantages of the proposed method over existing ones. Finally, the theoretical result is weak. It (Theorem 4.1) claims convergence to a continuous trajectory, but the property of this trajectory is unknown. Its convergence to the target distribution is important for it to be useful in applications. The non-uniqueness of the Kantorovich potential is also overlooked in the theoretical development.

**Additional Comments On Reviewer Discussion:**

The reviewers raise some questions on the results and presentations. The authors reply by modifying the paper, adding experiments in the paper, and adding clarifications in the response. Several reviewers are not convinced and keep their original evaluation of this work.

---

### Decision · Program_Chairs · 2025-01-22

Reject